# Access to Children's Perspectives?

**Anna Busk Rasmussen \* and Christina Haandbæk Schmidt \***

Department of Applied Science in Pedagogy and Society, University College Lillebælt, 5230 Odense, Denmark
\* Correspondence: abra@ucl.dk (A.B.R.); chhs1@ucl.dk (C.H.S.)

**Abstract:** Within the field of early childhood education, the Nordic model is characterised as being child-centred and holistic, based on children's participation, democracy, autonomy, and freedom. Despite a strong tradition of incorporating children's perspectives, research has identified it as a democratic problem that children continue to occupy a non-privileged position in which their voices are often unheard or disregarded in many contexts. Similarly, there is a tendency to apply adult-led methods, such as interviews, which can hinder the openness to children's diverse ways of communicating, which is not just through verbal expressions. In this article, we position ourselves within what we perceive as the second wave of research on child perspectives in which the research interest converges on exploring how children's perspectives are connected with the contexts in which children participate. Drawing on agential realism and an empirical example from a daycare centre, we demonstrate how children's perspectives emerge from and become entangled with pedagogues, ethics, spaces, materials and discourse. Thus, the question is not about gaining access to children's perspectives, but rather to be concerned with the interactions wherein children's perspectives can emerge. This involves a critical view of the structures and basic assumptions that manifest themselves in the daily life of daycare centres and which underlie, and can result in, a subordination of children and children's perspectives.

**Keywords:** children's perspectives; daycare; childism; ethics; agential realism

## 1. Introduction

Since the adoption of the UN Convention on the Rights of the Child in 1989, there has been a growing interest in incorporating children's perspectives into policy, childhood research, and pedagogical practices (Clark et al. 2005; Schwartz and Clark 2017; Kampmann et al. 2017; Warming 2019; Koch 2020; Gräfe and Englander 2022). In the Nordic countries there is a strong tradition of establishing child-centred policy models in the field of early childhood education, where children's participation, democracy, autonomy, and freedom are fundamental values. Collectively referred to as the Nordic model, this model is highlighted by organisations such as the OECD as examples of good pedagogical practices and serve as benchmarks for other countries (Hännikäinen 2016; Karila 2012). Although all the Nordic countries have ratified the Convention on the Rights of the Child, there is a difference in how the Convention is incorporated into the respective national legislations and what access children will have to appeal, if their rights have been violated. For instance, the Danish Daycare Act (2020) has been revised for the purpose of emphasising in its preamble that the pedagogical practice should be "based on a child perspective" (s. 7 of the Daycare Act). Similarly, the law stipulates that: "Daycare facilities must involve children in decision-making, share responsibility with them, and provide an understanding and experience of democracy" (s. 7(4) of the Daycare Act). The ministerial guideline states: "This means that children must be listened to, and their opinions taken seriously..." (the Ministry of Children and Social Affairs 2018, p. 8). The ratification of the UN Convention on the Rights of the Child, the Danish Daycare Act, and the ministerial guideline, entitle children in Denmark to be heard and involved in decisions that concern their lives, thus emphasising children's democratic rights. Nevertheless, the UN Convention on the Rights of

the Child is neither explicitly referred to in the revised Daycare Act nor in the 2018 national curriculum. Moreover, Denmark also has no specific children's ombudsman, such as is seen in Sweden, Finland, Norway, and Iceland; rather, this comes within the jurisdiction of the Parliamentary Ombudsman. Yet, a national council named "Børnerådet" (the Children's Council), established in 1994, operates independently of political and economic interests. This council aims at promoting the child perspective in public discourse and at raising awareness about legislators and others overlooking or neglecting children's interests (the Children's Council n.d.), but it does not have the authority to address children's appeals. A consolidated act, the children's act, implemented in 2024 (Child's Law 2024) with a view to establish "a child-friendly system", prioritises the legal security of the child (the Danish Authority of Social Services and Housing 2023).

Research has identified the fact that, in many contexts, children continue to occupy a non-privileged position in which their voices are frequently unheard or disregarded to be a democratic problem (Moosa-Mitha 2005; Warming 2018; Wall 2013, 2022). Warming emphasises how traditional understandings of child–adult dichotomies may potentially contribute to the suppression or silencing of children's voices rather than amplifying them (Warming 2020b). Tisdall (2015) underscores how this "silencing" emanates from the foundational adult-oriented systems into which we attempt to integrate children's participation. She particularly contends that this is evident from the children's challenging of normative perspectives within the "adult world". And (Sevón et al. 2024) argue that if children's initiatives and views are to be taken seriously, it is necessary to question adult dominance and apply a democratic and difference-centred perception of children that promotes equal participation.

A systematic review of international research aimed at capturing the voices of young children (Urbina-Garcia et al. 2022) reveals both a limited quantity of studies and a predominant use of adult-directed methods, such as structured interviews or focus group interviews. Similarly, the research indicates that questionnaires and interviews of older children are the most frequently applied methods when incorporating children's perspectives into the evaluation of pedagogical practices at daycare centres (Schwartz and Clark 2017). Koch (2020) also demonstrates how conversational approaches dominate pedagogical practices at daycare centres, with adults asking questions, listening attentively, and engaging with children's responses (Koch 2020).

Building on the Nordic tradition, where children are seen as active participants in their own life processes, researchers have developed methods to capture children's perspectives. Examples include observation-based interviews (Kampmann et al. 2017), doll-mediated interviews (Petersen 2017), future workshops (Alminde and Warming 2020), walking interviews (Rasmussen 2017), and the use of GoPro cameras (Hauge and Jørgensen 2019). These methods share the characteristic of being adult-initiated activities aimed at providing access to children's perspectives.

Both in research and in daily pedagogical practices, there is a tendency to utilise adult-directed, conversation-based approaches when accessing children's perspectives. This implies an entry into an adult-oriented system. Sundhall (2017) emphasises the naturalisation of adulthood, shedding light on how adult logics and norms, which, for instance, naturalise conversation as a kind of exchange and limit the possibility of investigating and understanding children's perspectives. Since conversation primarily relies on children's verbal expressions, the professionals may overlook that "children primarily communicate through their bodies and actions rather than verbally" (Koch and Jørgensen 2018, p. 112, our translation). Furthermore, the choice of "conversation" as the access point to children's perspectives denotes an understanding that children's perspectives are something they "have", and something to which we can gain access to by simply asking children what is on their minds.

In the context of Warming's conceptualisation of the child's perspective as encompassing children's rights and voices (Warming 2019, p. 67), it is evident, in addition to

being curious about children's voices, to take an interest in pedagogues as contributors to children's voices (Schmidt and Rasmussen 2022).

A current debate in childhood research challenges the perception of children's agency as something individually possessed, regardless of the contexts in which they are involved (Spyrou 2016; Esser et al. 2016; Alminde 2021). Through this, the limitations of affording children a voice and believing that they are true and authentic are emphasised. Instead, it is proposed that we must explore (and understand) children's perspectives as connected to the contexts in which children participate (Spyrou 2016; Warming 2019; Alminde 2020, 2021; Schmidt and Rasmussen 2022). This article critically examines the contexts within children's perspectives, rights, and voices that can emerge.

This identification of the problem field leads us to the following research questions:

Which challenges arise when pedagogues at daycare centres seek to promote children's democratic rights and perspectives?

## 2. Child Perspective in an Agential Realist Perspective

With this article, we position ourselves within the second wave of research on children's perspectives (Schmidt and Rasmussen 2022), where the research interest converges on the context, i.e., the environments in which children's perspectives can emerge. Whereas the first wave focused on understanding and conceptualising children's perspectives and developing methods to access them, our aim is to investigate how children's perspectives emerge in the daily interactions between children and pedagogues at daycare centres. For the purpose of being attuned to these interactions, we require a theoretical framework for encompassing not only the individuals—children and pedagogues—populating the context, but also the physical settings, discourse, and materials surrounding and shaping the context.

Building upon international research, we seek a framework that can assist us in challenging the assumption that children's perspectives are not something that "is" or that someone "possesses", but rather something that can "emerge" and is connected with specific contexts. Agential realism (Barad 2007; Taguchi 2010; Juelskjær 2019) can provide such a framework.

Agential realism is a part of a post-humanist and new materialist agenda (Juelskjær 2019, p. 13) that transcends linguistic as well as material turns, which assign agency to either language or materiality (Schmidt and Kortbek 2023). In agential realism, agency does not belong to subjects or objects (Juelskjær 2019). Instead, agency is attributed to "the in-between", i.e., the space that emerges from entanglements of humans, materiality as well as discourse (Taguchi 2010, p. 29). Thus, the focus shifts from human interactions to entanglements of both human, linguistic, and material actors.

Entanglement is part of Barad's 2007 theoretical framework through which we can understand and examine children's perspectives as something dynamic that arises in the encounter between children, pedagogues, rules, frameworks, discourse, and spatial arrangements, and not as something static that children "possess". When, in this article, we direct our attention towards the pedagogical practices of a Danish daycare centre, we can observe how pedagogues' actions involve specific understandings of the child–adult relationship and thus not only influence how they inquire but also how they engage in dialogues with children and which elements they incorporate. This, for example, becomes evident when pedagogues in a daycare seek access to children's perspectives in the playground of the institution. One pedagogue says:

> "We've also tried to give them (the children, ed.) cameras so they could take pictures of things they like to play with in the playground. So maybe we didn't get as much out of that as we thought we would. They took pictures of things like the swing, the slide and the sandbox, and that wasn't really what we were focusing on in this particular project, you could say" (Anonymous pedagogue from the empirical material 2021, our translation).

Even if the pedagogues do not use a verbal-conversation-based form, the example partly shows how the physical materials: the swing, the slide and the sandbox, play on the children's perspective which is expressed via the camera, and partly how children's perspectives are entangled with adult understandings of what is relevant to bring forward. In line with the international research, the example shows how the children's perspectives are rejected, as the pedagogues assess that "maybe we didn't get that much out of it" and "it wasn't really what we were focusing on". At the daycare centre, an adult-oriented system thus helps to silence the children's voices. In the following, we introduce the way in which we can use agential realism and its central concept of entanglements in the building of an apparatus (Barad 2007).

## 3. Apparatus

The consequence of agential realism is that entanglement does not merely constitute an analytical perspective for subsequent application to empirical material, as it should rather be considered as a research approach in which neither researcher nor the applied methods can be separated from knowledge (Barad 2007; Schmidt and Kortbek 2023). The studied phenomenon, children's perspectives, is intricately entangled with how and by whom it is studied and can be understood through what Barad refers to as the apparatus (Barad 2007).

Apparatus is not a noun but a verb, thus not denoting a research method but an act of doing. This doing occurs by way of simultaneous boundary-making and interrelated moves. Barad describes it as "boundary-making practices that are formative of matter and meaning, productive of, and part of the phenomena produced" (Barad 2007, p. 146). Agential realism entails a researcher position that does not observe from the outside but is involved and co-producing, both in the creation of empirical material and in its analyses (Schmidt and Rasmussen 2022). Hereby one of this article's authors, Anna, participated in the activities presented below through "participatory engagement" (Gunnarson 2018, p. 82, our translation), which is a further development of participatory observation that emphasises that the goal is to engage in the practice being investigated and not distance oneself from it.

In the following, we intend to demonstrate how we have constructed a research apparatus that is formative in the production of the phenomenon concerning children's perspectives within daycare practices. At the same time, this allows us to observe the phenomenon through boundary-making and interrelated moves through which we can identify the emergence of children's perspectives.

We acknowledge that conducting research involving children necessitates careful ethical consideration (Alminde 2020; Warming 2020a; Schmidt and Rasmussen 2022). Therefore, in addition to adhering to The Danish Code of Conduct and GDPR regulations, we approach each interaction in the field with sensitivity and empathy, prioritising the welfare of the children involved and their present state of well-being.

## 4. Professional Development Institutions

In this article, our apparatus is based on a local research and development concept known as "Faglige Udviklingsinstitutioner" (Professional Development Institutions). Within this setting, the two authors of the article participated in a project entitled "Children's Encounter with Children's Culture" 2022, together with representatives of the Social Education, two daycare centres, and two cultural institutions for children. The project proposed to investigate the various ways in which the involved actors understand and work with the phenomenon of children's culture. Additionally, it explored how, drawing inspiration from one another, they could develop practices within pedagogy at daycare centres, within dissemination in cultural institutions, and within teaching in the setting of social education (Applied Research in Pedagogy and Society 2022).

We apply this concept as a paradigmatic case (Warming 2022a; Flyvbjerg 2006) to illustrate how well-intentioned efforts to address children's democratic rights and perspectives

are framed within a specific project. This framing notwithstanding, ethical and pedagogical challenges will occur. By zooming in on a small segment of experienced childhood at a daycare centre, we demonstrate how the work with children's perspectives must be understood as inherently entangled with the underlying societal assumptions and structures present in the pedagogue's engagement with the children's perspectives.

The empirical material upon which this article is based originates from the project and includes observation notes, photos, as well as video and audio clips. From this material we have selected two empirical impacts from the same day but from two different locations. These two impacts form the empirical example which we will present in the following.

## 5. The Empirical Example

As an element in the project "Children's Encounters with Child Culture", one of the article's two authors, Anna, participated in a sub-project together with a researcher and two daycare pedagogues. In the selected example, the involved pedagogues, Lars and Ulla, the researcher Ida, and Anna visited a children's theatre together with a group of eight 5-year-old children for the purpose of watching the performance "Lille Myr".

The first impact of the paradigmatic case takes place at the theatre.

It is dark in the large room we enter. Only a small scene is lit up. Both children and adults sit down on rows of spectators that rise like a staircase away from the stage. Anna sits in the middle and one of the children, Mary aged 5, sits on her lap.

On stage is a soft scenography in a "water/lake universe" with large leaves, seaweeds and flowers in fabric and knitted or crocheted yarn. At the back is a large screen, which forms the backdrop for the scenography of the performance. Sound and music, together with changing light, contribute to creating the moods of the performance. The actor leads the action forward and controls the puppet "Lille Myr". At one point, the actor swims in the lake and makes swimming movements that cast shadows on the wall. Instead of looking at the scene, the girl on Anna's lap, Mary, has her attention captured by the shadows dancing on the wall outside the stage. Mary softly notes this to Anna. Anna responds by looking the same way as Mary.

The second impact of the case takes place in the workshop of the daycare centre after the performance. In the workshop there are various hobby materials and tables, as well as chairs are at children's level. The participants are Ulla and Lars, Ida and Anna, and the eight children who attended the performance. The professional's starting point is the mutual understanding that the agenda is to allow the children an opportunity to engage with their experience in the theatre and in their habitual surroundings.

Initially Ulla, a mischievous glint in her eye, addresses Ida, saying, "I've just come up with a fantastic plan". "That sounds exciting", says Ida. Ulla and Lars stand up in front of the children, displaying the two posters from the performance. They ask the children, "What did we just experience at the theatre?" "A tadpole", one child answers. "Yes, and what did the tadpole turn into?" Ulla asks. "A frog", another child replies. "Yes", says the pedagogue, Ulla, smiling and nodding.

Anna urges the children to try to draw what they witnessed at the theatre on the same day. She asks, "What did you see and hear inside the theatre?" "What did it look like?" she asks. "It was dark", says one girl. "Can I have the dark colour?" a boy quickly asks. Anna continues, "What did you think of Lille Myr? Try to draw that". The children immediately begin, absorbed in their work. One child begins by covering the entire paper in black, as the darkness in the auditorium loomed large in several children's experience of Lille Myr. Others draw leaves or tadpoles. One draws light at the entrance and darkness in the auditorium.

Several children say they cannot remember what they experienced at the theatre. Anna asks a child who sits still, looking at the others painting: "What do you want to draw?". The child lifts her shoulders and sighs. Another child at her table says: "but you have to draw, what was at the theatre". The child wants to see a photo, and Anna shows her a picture on her phone. The child begins to paint.

Another child sits thinking for a moment, then begins to draw leaves of different colours. When the children have completed their drawings, the adults help them to write what they have drawn on their pictures.

Finally, before the children leave for the playground, Anna asks them a question. She stands up saying, "Ida and I will visit the kindergarten again tomorrow. Today, we watched Lille Myr and drew what we experienced. We would like to ask you if you would like to continue playing with Lille Myr tomorrow?" The children are silent for a while, thinking. Some of them look at Anna with a puzzled expression. The adults are also quiet. Suddenly, a child exclaims, "We can make our own Lille Myr—out of popsicle sticks". "Yes", says Anna, "We can do that". "Then we can create our own theatre", the child continues. The other children agree; they would like to be a part of that.

The pedagogues immediately begin to rummage for popsicle sticks and other things they can use the next day. Meanwhile, some children run out into the playground, whereas others complete their drawings. "We can also use fabric and stuff it with cotton", Lars suggests. Anna proposes a cardboard body glued to a popsicle stick so that the children can control the tadpole themselves. "Shall we make a template then?" Ulla suggests. "No", says Anna, "The children can draw the body themselves and then cut it out". "Then we'll have lots of different tadpoles", Lars continues. "Oh yes!" says Ulla, laughing. "That's true". "We practice", she continues. "And we help one another", Anna smiles.

## 6. Analytical Readings

For the purpose of anchoring the research ambition, we draw inspiration from analytical reading strategies (Søndergaard 2018; Khawaja 2018). We understand analytical reading strategies as a means to connect our research questions with an analysis of our empirical material. This connection is established by reading the empirical material with different conceptual foci in order to highlight various elements in the entanglements that occur when children's perspectives emerge (Khawaja 2018, p. 167).

Our conceptual foci are inspired by Childhood Prism research (Warming 2022b), which represents an approach to research on children's lived lives that is rooted in the childhood studies tradition, also acknowledging the fact that childhood is a social phenomenon (Warming 2022b). Based on this inspiration, we can get close to children's lived everyday lives at a daycare centre while, at the same time, challenging the structures, norms and understandings that manifest themselves in daily life at the institution, and that underlie and can result in a subordination of children and children's perspectives.

Thus, the first analytical reading is based on the concept of "child". The choice of this conceptual focus underlines the ambition to zoom in on the concrete children and their lived lives at a daycare centre. The focal point is when and how children's perspectives emerge. This focus is founded in childhood studies, which occurred during the 1980s, primarily addressing the fact that we must understand children as social actors who create their circumstances as well as being created by them (James et al. 1998). Unlike the predominant developmental psychology, this new paradigm argued that there is not just one natural or universal childhood but multiple childhoods and experiences of being a child, and it supports the idea that children should be understood as "beings". This is in line with the Danish tradition of democracy at daycare centres and the view on children as active participants in their own lives (Kampmann and Prins 2022). To help us approach the context in which children's perspectives can emerge, we draw on Davies' 2014 concept of "emergent listening". Through this, Davies emphasises the importance of listening for the not-yet-known. Emergent listening requires an openness to what we do not yet know, and it entails our avoidance of understanding what is being said, considering what we already know or suspect. Thus, it is not just an extension of our usual ways of listening, it is also about setting aside the adult's understanding of the world (Davies 2014).

In the empirical example there was a pronounced significance of allowing the children's perspectives to be expressed and, thus, this reading can contribute with empirical analyses of contexts in which children's perspectives (can) emerge.

The second analytical reading is based on the concept of 'childism'. The phenomenon of childism combines research and political activism and aims to combat child discrimination. The normative benchmark of childism is social justice and equality for children (Warming 2020a, p. 69). As childism aims to criticise and reconstruct fundamental normative assumptions (Wall 2022), this focus allows us to zoom out and take our reading against the first conceptual focus in order to encapsulate such perspectives as we failed to capture on our first attempt. Childism focuses on how traditional understandings of child–adult dichotomies can potentially silence children's voices rather than bring them alive (Warming 2020b). Thus, this reading enables a criticism of the othering of children through a critical analysis of the generative mechanisms that shape the lives of children (and of adults) and their opportunities to assert their perspectives.

Using an agential realistic impetus and its concept of entanglement, the analysis will unfold alternating between the two conceptual foci. The ambition of this move is to show how the concrete children, as well as the concept of childhood, will always be entangled in the contexts within which children's perspectives emerge. This will offer a nuanced perspective on the challenges entangled with pedagogical practices in which pedagogues at daycare centres seek to promote children's democratic rights and perspectives.

## 7. The Open Invitation

The first empirical impact is the setting and invitation to the children's work with their experience at the theatre. At first, we zoom in on how and when children's perspectives emerge. The pedagogues strive to invite the children into their own experiences, doing so openly and equally:

> Ulla and Lars stand up in front of the children, displaying the two posters from the performance. They ask the children, "What did we just experience at the theatre?" "A tadpole", one child answers. "Yes, and what did the tadpole turn into?" Ulla asks. "A frog", another child replies. "Yes", says the pedagogue, Ulla, smiling and nodding.

Ulla and Lars are apparently concerned with the idea of reminding the children of their experience from the theatre as they take advantage of the opportunity to work openly on the performance. The poster from the performance apparently works as material for engaging with the children's experience, allowing them to connect with their perspectives. The poster is entangled with the discourse of children's own premises. Next, Anna encourages the children to connect with their experience through their senses:

> Anna urges the children to try to draw what they witnessed at the theatre on the same day. She asks, "What did you see and hear inside the theatre?" "What did it look like?" she asks. "It was dark", says one girl. "Can I have the dark colour?" a boy quickly asks. Anna continues, "What did you think of Lille Myr? Try to draw that".

In this example, by asking open-ended questions and offering the children different materials, Anna strives to listen to what is emerging on their minds in their interpretations of the performance. There are no right answers, but opportunities to recall experiences, experiment with materials and express themselves aesthetically. This approach provides the children with a way to express themselves by other means than verbally and thus supports the fact that children communicate in diverse ways. By working aesthetically, the adults aim at allowing all children to be heard; including the ones who do not have the opportunity to express themselves verbally. Thus, the pedagogues connect with the children's ethical and democratic right to be heard. The present discourse of children's own premises and experiences in their own rights is rooted in a stated intention, among pedagogues, researchers and educators alike, to invite the children into the processing of the theatrical performance without defining, or patenting, their experience. At the pre-planning meeting, pedagogues, educators and researchers agree that:

"We want to explore how we can give the children the opportunity to enter into dialogues with the experience—on their own terms. We will strive to invite the children into the production of the performance, openly and without defining or patenting their experience" (Field notes from planning meeting 2022)

This corresponds well with the idea in childhood studies that children must be understood as beings in their own rights and, likewise, it is in line with the regulatory requirement that children should be allowed to be heard. It is moreover in line with the Nordic tradition in which children's participation, democracy, autonomy, and freedom are fundamental values. This partly forms the discourse, present in the empirical example. This discourse becomes entangled with the materials present. The choice of the workshop, furnished with tables and chairs at the children's level and creating a setting in which the children can participate undisturbed, supports the idea of equal and democratic room for the children's interpretations of the performance. At first glance it seems that the adults succeed in inviting the children to engage with the experience on their own terms.

Allowing ourselves to zoom out with "childism", the ambition of the second reading is to look at the distinction between children and adults as an expression of a social structure and to take a critical look at the, often dichotomous, understandings of the child–adult relationship that is expressed in pedagogical practice. With this theoretical lens, we can zoom out and spot the poster as a materiality that emphasises the truth about the performance "Lille Myr", and thereby defines what is the "right" answer to the pedagogues unspoken question. In continuation of the presentation of the poster, the pedagogue asks the children questions about the tadpole who turned into a frog. Through a childist lens, we understand this conversation as an expression of an unequal social structure. Although the pedagogues have the best intentions of letting the children engage with their experience on their own terms, they nonetheless become entangled with a basic assumption about children as beings who are not (sufficiently) competent to know their own experience and, hence, must be led on the right track. The pedagogue determines what is important: "What did the tadpole turn into?" and the children responds. When they give the right answer the pedagogue smiles and nods. She offers a position for the children as the ones who should look for correct answers, when the adult asks questions. Despite good intentions to the contrary, an adult-oriented system can thus potentially dampen children's voices and limit their democratic rights and perspectives.

With the agential realistic position, we, as researchers, must be aware that the knowledge we produce cannot be understood independently of our presence in the project and in the specific workshop. Our bodies interact with and co-produce the discourse and adult positions that exist in spaces, materials and pedagogic approach. When, in the following empirical impact, Anna appears as an adult and as someone who instructs the children to draw on their experiences, she helps to produce adult positions and create participation opportunities for the children.

## 8. The Experience

In the next empirical impact, we zoom in on the concept of "child", on the children's expressions through their paintings of the theatrical performance. We use this lens to get close to the children and their engagements with adults and materials. It seems that the children draw what fills their minds:

One child begins by covering the entire paper in black, as the darkness in the auditorium loomed large in several children's experience of Lille Myr. Others draw leaves or tadpoles. One draws light at the entrance and darkness in the auditorium.

Zooming in on the concrete children, we are able to unveil how the adults have made a space for the children to express themselves in diverse ways and hence, support multiple experiences of being a child. With the concept of "emergent listening", we can emphasise how this creates an opportunity for the pedagogues to listen emergently. If the

pedagogues succeed in disregarding their own understanding of the theatre, this can create an opportunity for making space for the not-yet-known.

> Several children say they cannot remember what they experienced at the theatre. Anna asks a child who sits still, looking at the others painting: "What do you want to draw?". The child lifts her shoulders and sighs. Another child at her table says: "but you have to draw, what was at the theatre". The child wants to see a photo, and Anna shows her a picture on her phone. The child begins to paint.

The photo works as a material that engages with the experience. The photo helps to elicit the hesitant child's action, as the child begins to draw.

But when we zoom out and read the sequence with "childism", we become aware of the way traditional understandings of child–adult dichotomies are present in the discourse offered. The idea that the adults know best, and the children know less, is exposed through the poster material, and the underlining of the theme of the performance: the tadpole who turned into a frog, guiding the children to focus on what happened on stage during the performance and not on all the other impressions they had, such as Mary's fascination of the shadows. This is revealed when another child talks through the discourse offered by the adults and urges the child to draw what was at the theatre. If we look back at the invitation, this is also the case when Anna, seemingly, invites the children openly and without the right answers, but still concludes the invitation with the words: "What do you think of Lille Myr? Try to draw that", thus steering the children's attention to the actual theatre. And in addition to this, she offers the child, who does not know what to draw, a photo of the theatre. From an adult logic perspective, it is obvious that the experience concerns what happened on stage, but by transferring this logic onto the children, we overlook the opportunity for something different and (perhaps) more important to emerge which actually tells us something that we did not know already. In this particular case, Mary, the girl on Anna's lap, was so fascinated by the shadows on the wall during the performance, that she never really looked at the stage. We can only guess at what she would have shown, if we had set our adult logic aside and listened more emergently.

## 9. Discussion and Final Conclusions

The challenges that arise when pedagogues at daycare centres seek to promote children's democratic rights and perspectives are not necessarily a question about gaining access to children's perspectives, but rather to be concerned with the interactions wherein children's perspectives can emerge. This involves a critical view of the structures and basic assumptions that manifest themselves in the daily life of the daycare centres and that underlie, and can result in, a subordination of children and children's perspectives. The central argument of this article is that listening to young children's voices not only requires an understanding of listening as an ethical relationship based on respect for difference and for the other. It also calls for a constant questioning of what comes to life as meaningful and how this entangles with children's possibilities to think and act. This requires pedagogues who dare to be curious about their common practice and care to question whether a template is necessary or not, and strive to make room for the not-yet-known.

Drawing on agential realism and a paradigmatic case, we have demonstrated how children's perspectives emerge from and become entangled with pedagogues, ethics, spaces, materials and discourse. Based on an analytical reading strategy, with two conceptual foci, we have highlighted various elements in the entanglements. Zooming in on the empirical example with a theoretical focus on 'child' and a childhood study approach, we have exposed how the Nordic understanding of the child is present in the intentions and ways in which the pedagogues organise and enable the children's interpretations of the theatrical performance. We see how children's voices emerge from and are entangled with the way in which the workshop is set up, the choice of aesthetic materials, and the ways in which the pedagogues ask open-ended questions that enable democratic participation. When zooming out, within the concept of "childism", we uncover the fact that, despite the good intentions of emergent listening together with conscious planning and framing on the part

of the adults, we sometimes end up silencing children's perspectives. Asking questions and showing pictures based on normative assumptions and child–adult dichotomies, we fail to capture the children's own experiences.

Through this article, we have made an effort to visualise the complexity, and create a more nuanced understanding, of children's perspectives. Initially, reading the empirical material in the light of the concept of "child" and, next, in the light of the concept of "childism", we demonstrate how children's perspectives become entangled in boundary-making practices. Similarly, we demonstrate the presence of a need to question the usual, often dichotomous, ways in which we understand the child–adult relationship at daycare centres and in society at large. This calls for an increased attention to the chaotic, complex, and multifaceted entanglements in the emergence of children's perspectives.

**Author Contributions:** Conceptualization, A.B.R. and C.H.S.; methodology, A.B.R. and C.H.S.; software, A.B.R.; validation, A.B.R. and C.H.S.; formal analysis, A.B.R. and C.H.S.; investigation, A.B.R.; resources, A.B.R.; data curation, A.B.R.; writing—original draft preparation, A.B.R. and C.H.S.; writing—review and editing, A.B.R. and C.H.S.; visualization, no visualization; supervision, C.H.S.; project administration, C.H.S. All authors have read and agreed to the published version of the manuscript.

**Funding:** No external funding has been received for this project.

**Institutional Review Board Statement:** Ethical approval was waived for this study. In accordance with a statement from Poul Skov Dahl, Director of Research at UCL, University College, Denmark, the study is compliant with relevant Danish and international standards and guidelines for research ethics. Please note that the project complies with Danish law, since no ethical approval is demanded for this type of research project. Accordingly, no Danish authority exists, where approval could be obtained.

**Informed Consent Statement:** Informed consent was obtained from all subjects involved in the study.

**Data Availability Statement:** The data presented in this study are available on request from the corresponding author due to privacy issues.

**Conflicts of Interest:** The author declares no conflict of interest.

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
