# Peer review of "Access to Children’s Perspectives?"

_socsci, doi:10.3390/socsci13030163_

Round 1

Reviewer 1 Report

Comments and Suggestions for Authors

The goal of this paper is to explore a new perspective and method for revealing young children’s experiences within early childhood care and education settings and especially how context can affect children's perspectives. The orienting research question stated is, “Which challenges arise when pedagogues at daycare centres seek to promote children's democratic rights and perspectives?” In their review of previous research focused on children's perspectives, they find that traditional methods usually emphasized adult-led verbal conversations, which provide limited access to children’s perspectives. The authors chose to use the theoretical perspective of agential realism, with its central concept of “entanglements” to explore this research question.

The authors focused their study on a specific example, that of two childcare pedagogues’ efforts to gain access to children’s perspectives following attendance at a children’s theatre performance. The researcher observed the children’s and pedagogues’ interactions, using written notes, photographs, video and audio recordings. In the observed example, the pedagogues lead a discussion of the recent theatre performance and make suggestions to the children about what they might draw or paint to represent their experiences. The researchers analyze this event using two different perspectives: focusing first on the concept of the “child”, and then with a separate second reading of the data from the perspective of “childism.” Using these two research perspectives, they point out distinctly different ways of viewing these adult-child interactions. From the “child” perspective, the children are given opportunities to express their individual responses to the adults’ open-ended questions, which can been seen as not prescriptive, but instead encouraging of individual expression.  From the “childism” perspective, the children are seen as constrained in their expressions by the adults’ particular directions and suggestions. Thus, the authors illustrate the “entanglements” of children’s perspectives in the context of their interactions with their teachers.

While this short paper is well-conceived and clear in its description of the theoretical perspective and observation method, the observations summarized are few, and somewhat sparce.  I believe if more examples of the observed adult-child interactions were provided in this paper, the analyses and conclusions could be strengthened.

The authors state in their discussion that they have attempted to show “the complexity of and a more nuanced understanding of children’s perspectives.” I believe they have opened the door for readers to think about at least one aspect of that complexity. They make a very important point that the structures that adults impose when guiding children’s activities (i.e., “boundary making”) foreclose revelation and understanding of at least some of children's perspectives. A richer report of the observed interactions in this post-theatre activity would almost certainly strengthen their case, plus provide the reader with more ideas about how to engage children respectfully and increase understand their perspectives.

Reviewer 2 Report

Comments and Suggestions for Authors

An interesting perspective. However, not really accessible language use. It took some effort to read the article from beginning to end. I wonder how many people will read the article.

I wonder how old the children in the empirical part are. I can't imagine that children in a daycare center understand and use the kind of language presented in this article. Neither that they can draw the things mentioned here. This can't be an average daycare center.

Reviewer 3 Report

Comments and Suggestions for Authors

Access to Children´s Perspectives?

This article focuses on a current and relevant problem in the field of early childhood education, democracy and children rights. I consider that this article could be accepted with minor revision. I indicated some of theses minor changes in the file which I am sending.

All the best.
